# Low-Frequency Vibration Sensor with Dual-Fiber Fabry–Perot Interferometer Using a Low-Coherence LED

Mu-Chun Wang [1,*], Shou-Yen Chao [1], Chun-Yeon Lin [2,*], Cheng-Hsun-Tony Chang [1] and Wen-How Lan [3]

1   Department of Electronic Engineering, Minghsin University of Science and Technology, Hsinchu 30401, Taiwan; shouyen@must.edu.tw (S.-Y.C.); chtchang@must.edu.tw (C.-H.-T.C.)
2   Department of Mechanical Engineering, National Taiwan University, Taipei 106319, Taiwan
3   Department of Electrical Engineering, National University of Kaohsiung, Kaohsiung 81148, Taiwan; whlan@nuk.edu.tw
*   Correspondence: mucwang@must.edu.tw (M.-C.W.); chunyeonlin@ntu.edu.tw (C.-Y.L.); Tel.: +886-3-5593142 (M.-C.W.); +886-2-33662725 (C.-Y.L.)

**Abstract:** In this paper, we propose a dual-fiberoptic Fabry–Perot interferometer (FFPI) sensing system integrated with a low-cost and low-coherence light-emitting diode (LED) as a light source to detect dynamic vibration caused by acoustic waves with a cut-off frequency of 200 Hz. When the acoustic signals are applied, the sensing FFPI on a Styrofoam sheet provides the function of partially transforming the longitudinal energy as the transverse energy generates a phase shift in the sensing FFPI cavity. The light reflected from the sensor is demodulated by the reference FFPI to extract the measurand. The low-power (sub-nW) optical signals are transferred into electrical signals, processed by a designed optical receiver, and recorded for data analysis.

**Keywords:** Fabry–Perot interferometer; vibration; acoustic wave; sensing system; LED; coherence





## 1. Introduction

Refractometry optics are widely used in industry, biology, semiconductor fields, etc., especially with fiberoptic architecture [1–9]. The applications of fiber optics were first introduced more than 70 years ago, such as in communication, parameter extraction, image transmission, and sensing [10–14]. With respect to sensing, given the view of mass production and precision, novel fiber Bragg grating (FBG) technology [15–22] is preferable to primary optic interferometers [23–29], despite its high cost and complicated manufacturing process. Given their operational convenience, system complexity, and cost-effectiveness, fiberoptic interferometers are valuable for small-scale or pilot experiments [30–35].

Despite studies exposing the vibration effect [36–38] with the aim of inducing a mirror disturbance in the sensed interferometer via acoustic vibration, indirectly changing the cavity length and impacting optical transmission is another interesting topic. Due to this phenomenon, the desired sensing parameter(s), such as acoustics, mechanical strain or stress, thermal stress, or electrical stress, can be extracted from the disturbance source. These precisely reflected consequences are beneficial for system designers in terms of accurately monitoring and post data processing. Refractometry acoustic sensors have been used since they were first proposed to detect pressure by Bucaro et al. in 1977 [39–44]. Owing to benefits such as low cost, accuracy, high reliability, immunity to electromagnetic interference, ability to operate in a broad range of environments [45,46], high sensitivity, simplicity, compactness, and the potential for multiplexing [47], single-fiber Fabry–Perot interferometers (FFPIs) with a laser diode light source have been widely used to sense temperature [48,49], pressure [50–52], and acoustic wave amplitudes [41,42]. However, the cost of such devices for single-mode operation is relatively high, and a Faraday isolator is necessary to eliminate feedback effects. The operating temperature range is limited, and thermal-induced shifts in the emission wavelength of the laser are needed to compensate.

Motivated by the need to design and fabricate a low-cost FFPI sensing system, we used a communication light emitting diode (LED) as a light source. The proposed system was designed to function as a sensing system with a laser light source [40] in dynamic operation. A dual FFPI sensing system integrated with a 850 nm low-cost communication LED was utilized to sense acoustic vibration signals. This sensing system involves FFPI sensing and a reference FFPI.

## 2. Sensing Principles of a Dual FFPI and Experimental Setup

Exposing the operating principles of a dual FFPI is essential to provide accurate information to establishing a feasible experimental setup.

### 2.1. Basic Principles of a Dual FFPI

The Fabry–Perot interferometer constructed by Charles Fabry and Alfred Perot in 1898 plays an important role in modern optics. Over the years, this concept has found a broad spectrum of applications. In addition to being a spectroscopic device with extremely high resolving power, it can serve as a basic laser resonant cavity or an optical oscillator. With the development of optical fiber, this interferometer is also utilized as a highly sensitive sensor head. A schematic diagram of a single FFPI is shown in Figure 1.

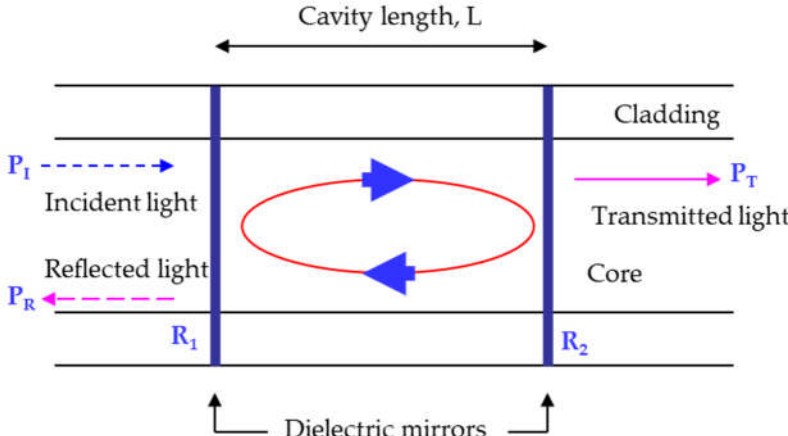

**Figure 1.** Illustration of a single FFPI with interior mirrors.

By ignoring the optical loss in [51], the power reflected from a single FFPI can be expressed as:

$$P_R = \frac{R_1 + R_2 - 2(R_1 R_2)^{1/2} \cos \varphi}{1 + R_1 R_2 - 2(R_1 R_2)^{1/2} \cos \varphi} P_I \tag{1}$$

where $R_1$ and $R_2$ are the reflectances of two mirrors in a single FFPI, $P_I$ is the total power incident into this single FFPI, and $\varphi$ is the roundtrip phase shift. $\varphi$ is also dependent on the peak wavelength of the light source ($\lambda$), the refractive index of the cavity fiber medium $n$ (=1.46 for silica), and the unperturbed cavity length of the interferometer ($L$) according to:

$$\varphi = \frac{4\pi}{\lambda} \cdot (nL) = \frac{4\pi \, \nu}{c} \cdot L_o \tag{2}$$

where $\nu$ is the optical frequency, $c$ is the optical speed, and $L_o$ is $nL$.

As the optical loss in the cavity is temporarily omitted due to the minimal loss and $R_1 = R_2$ are equal to $R$, Equation (1) and the transmitted power $P_T$ can be simplified as:

$$P_R = \frac{2R - 2R \cos \varphi}{1 + R^2 - 2R \cos \varphi} P_I \tag{3}$$

$$P_T = \frac{P_I \cdot (1-R)^2}{1 + R^2 - 2R\cos\varphi} \tag{4}$$

If $R << 1$, Equation (3) can be expressed as:

$$P_R = 2P_I \cdot R \cdot (1 - \cos\varphi) \tag{5}$$

A dual FFPI sensor [30,49] with a diode light source was explored to sense the continuous-wave temperature variation. Using this characteristic, the applications are expanded to dynamic sensing. As shown in Figure 2, the FFPI sensing system is composed of a sensing FFPI, a reference FFPI, an LED driver, a 50/50 fiber coupler, a feedback circuit, and an optical receiver. This working principle is similar to an electronic circuit design [53], which works at the operating point with direct-current bias and then processes the small dynamic signal via amplification or filtering.

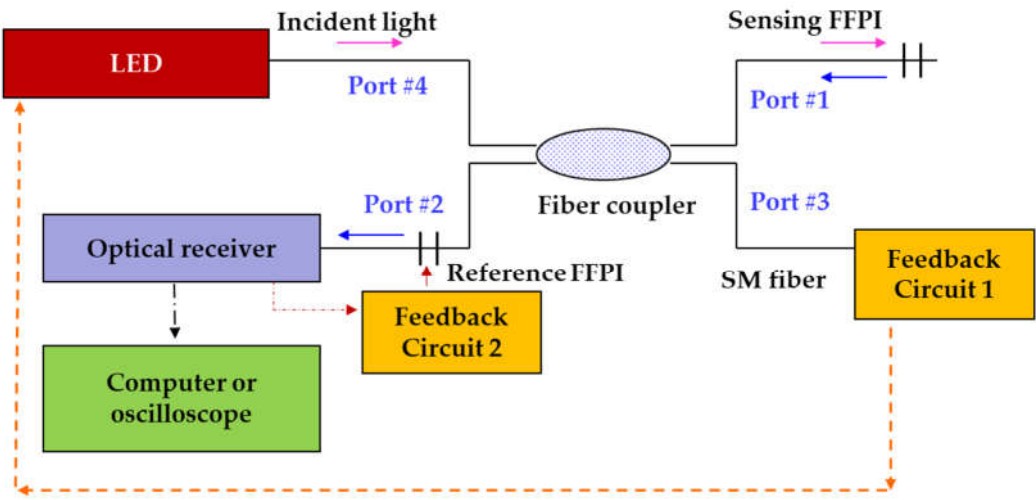

**Figure 2.** Schematic functional block diagram of a dual FFPI [30].

The reflected power from sensing a FFPI ($P_{Rs}$) is treated as incident power at a reference FFPI site, $P_{Ir}$. $P_{Ir} = \alpha P_{Rs}$ represents the optical loss between sensing and reference FFPIs, where $\alpha$ is constant.

The modulation effect shows the power reflected from the reference FFPI, represented as:

$$P_{Rr} = 4\alpha\, R_s\, R_r P_{Is}\{1 + 0.5\cos\Delta\varphi_p \exp[-(a\Delta L_o)^2]\} \tag{6}$$

$$a = \frac{\pi\Delta\nu}{c \cdot \sqrt{\ln 2}} \tag{7}$$

where $R_s$ and $R_r$ are the reflectances of the mirrors in the sensing and reference FFPIs, respectively; the constant $a$ is related to the light source spectral width ($\Delta\nu$); $\Delta\varphi_p$ is the phase shift due to the perturbation applied to the sensing interferometer; and $\Delta L_o$ is the optical path length mismatch between reference and sensing FFPIs.

Similarly, the transmitted power from the reference FFPI ($P_{Tr}$) is approximated by:

$$P_{Tr} \approx 2\alpha^2\, R_s P_I\{1 - R_r \cos\Delta\varphi_p \exp[-(a\Delta L_o)^2]\} \tag{8}$$

*2.2. Inner Mirror Fabrication and Experimental Setup*

Using a sputtering system, a thin titanium oxide (TiO$_2$) layer (~1000 Å) was deposited as a reflected mirror in an FFPI. These fibers are moved on a cleaver with an adjustable one-dimensional stage, shown in Figure 3a. Precision cleaving of both fibers is simultaneously employed. A microscope is applied to inspect the alignment and the cavity lengths during this cleaving. There is an internal ruler in this microscope. The smallest division in this ruler is about 15 μm for a maximum magnification $\approx 1000\times$. One scale marker in the ruler

is about 4 μm. In the beginning, the smallest magnification is used to roughly align both dielectric mirrors, which are light blue under the microscope. The joint between the coated fiber and the uncoated fiber is clear and light blue in color. There is also a ruler in the cleaver to inspect the cavity length. In this work, a 12-mm-long cavity for internal-mirror FFPIs was fabricated, exhibiting an adequate reflectance with precise fiber cleaving and splicing. The dielectric mirror was moved to the number 12 position in the cleaver with a microscope, as shown in Figure 3b. Ultimately, the desired cavity length was achieved by moving a diamond blade in the cleaver. The same procedure was employed to achieve identical cavity lengths for both fibers simultaneously. When these two fibers with dielectric mirrors were moved to approximately the number 12 position in the cleaver, one of the fibers was fastened on a 1D stage with tape. The difference between these two cavity lengths (less than 5 μm) was obtained with the highest magnification and an adjustable stage. The smallest division is ≤10 μm. If the desired cavity length is not obtained with a cleaver, a micro polish machine can be applied to adjust the length of one fiber as needed.

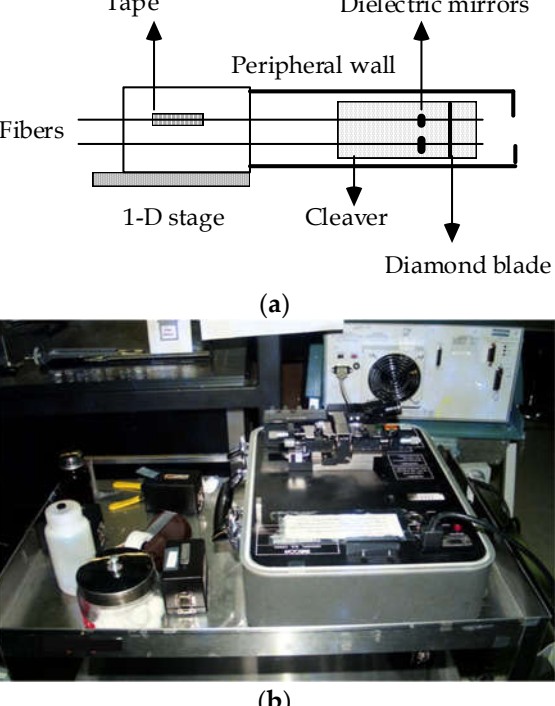

(**a**)

(**b**)

**Figure 3.** (**a**) Top view of the alignment of both fibers with dielectric mirrors on a cleaver and an adjustable 1D stage. (**b**) Contour of a splicer.

Moreover, it is necessary to test the spectral output of the applied light source to eliminate the error source in the sensing view. The monochromator is utilized to verify the performance. In the experimental setup, an LED is fixed on a three-dimension positioner for localization. Additionally, by tuning the positioner and an infrared-sensitive card, the LED can be aligned at a focus point on the left-hand side of a fixed lens. Because the LED was at the focus point, collimated light was incident upon a Jarrell-Ash monochromator. A chopper with the frequency set to 1 kHz was located between the lens and the monochromator. An electrical signal from the chopper was directed to a lock-in amplifier as a reference signal. Light from the LED passing through the chopper entered the monochromator and was diffracted by a grating, which can be rotated with a stepping motor. A power meter collected light from the output slit of the monochromator. The electrical signal from the power meter entered the lock-in amplifier, the output of which directed the y-axis of the plotter. When the grating drive was operating, the LED spectrum was traced by the x-y plotter.

The experimental results, as shown in Figure 4, demonstrate that the spectral distribution of this LED is close to a Gaussian shape. Therefore, as in [54], the coherence length ($L_c$) is expressed as:

$$L_c = 0.664 \cdot \frac{\lambda^2}{\Delta\lambda} \tag{9}$$

where $\lambda$ is the peak response wavelength, and $\Delta\lambda$ is the full width at half maximum (FWHM) of the spectrum.

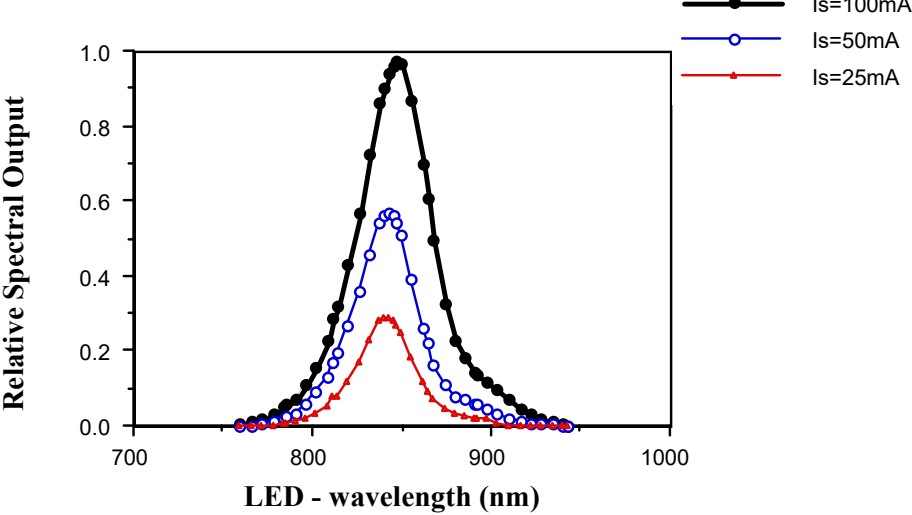

**Figure 4.** Relative spectral output vs. wavelength of the tested diode measured with a monochromator. There are three close Gaussian shapes at the forward currents: $I_s$ = 25, 50, and 100 mA [30].

According to the data sheet in [54] and the measurand, $\Delta\lambda$ is approximately 60 nm. Thus, the coherence length ($L_c$) in this light source, as determined by Equation (9), is approximately 8 μm.

The sensing FFPI was placed on a Peltier device or thermoelectric cooler (TEC Marlow SP1546T). This cooler exhibits an excellent temperature linearity characteristic with a forcing current of 0 to 40 °C. The TEC in the reference FFPI was treated as an optical path-length change compensator to maintain the optimal system operating point. To verify the surface temperature of the TEC, a K-type thermoresistor was also placed on top of the TEC [30]. The variation data of the thermoresistors in the reference FFPI were automatically recorded by a data acquisition system connected to a personal computer. As the mismatch length of these cavities decreased to less than *L* or zero, the distortion of the sensing signal in the system was expectedly improved. An acoustic speaker utilized as a vibration-wave source, as depicted in Figure 5, was suspended on a sensing FFPI with a Styrofoam sheet as an acoustic compressor to stress the sheet and produce a disturbance on the sensing FFPI. Owing to the characteristics of a Styrofoam sheet, it partially transforms the longitudinal energy as the transverse energy and indirectly impact the phase change of the sensing FFPI.

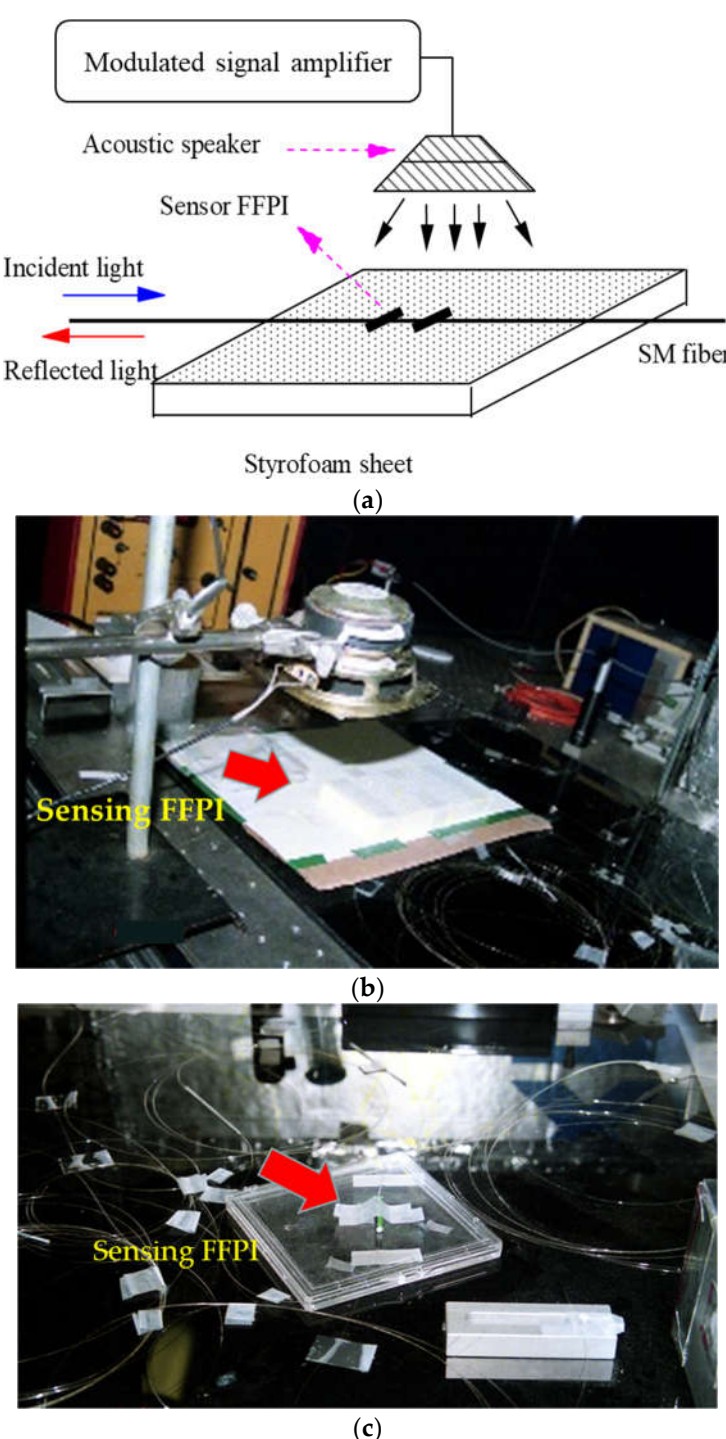

**Figure 5.** Experimental arrangement to test the acoustic response of sensors with an acoustic speaker, a sensor FFPI, and a Styrofoam sheet: (**a**) schematic illustration, (**b**) physical setup, and (**c**) sensing FFPI outside the acoustic setup.

## 3. Results and Discussion

Using this LED light source, the operating stability is also an essential factor to ensure sensing performance. The stability consequences of this diode light source in short- and long-term tests sensed by an optical detector are exhibited in Figure 6. In general, the maximum and minimum peaks are 0.7015 V and 0.698 V, respectively, for the short-term test. The maximum value is 0.702V, with a minimum value of 0.694 V for the short-term test. This tested diode demonstrates the suitable stability of the expectation: below 1%

at room temperature. These experiments were repeated in triplicate, with a performance variation of less than 1%. Then, a simulation was executed to achieve suitable reflectance, as shown in Figure 7 [30]. Here, the experimental reflectance of a single FFPI of $R \approx 0.1$ was adopted.

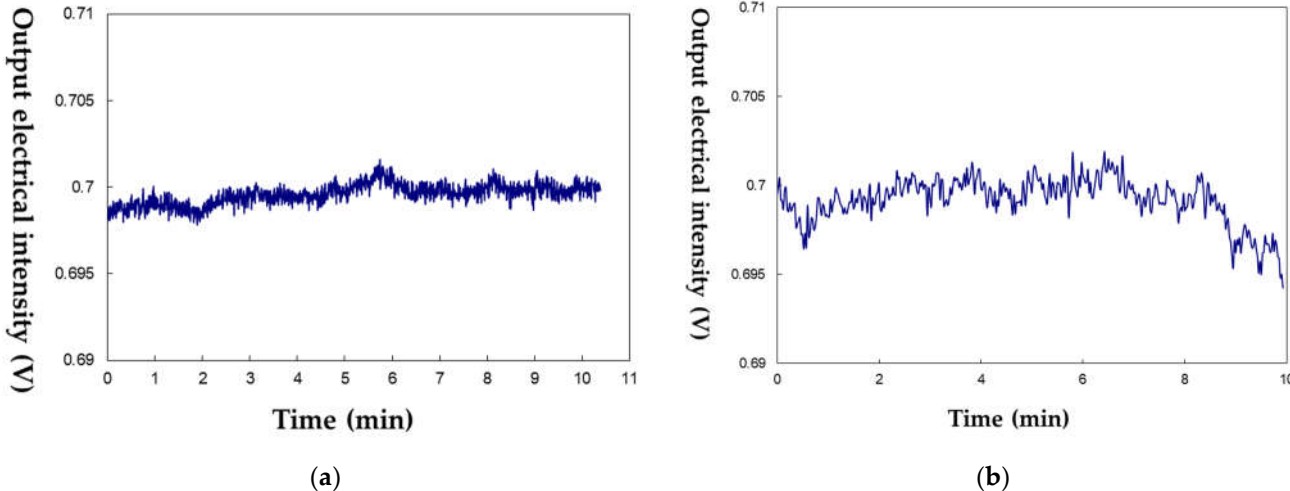

(a)　　　　　　　　　　　　　　　　　　　　　　　　　　(b)

**Figure 6.** Optical power stability tests for the experimental light source outputted with relative electrical voltage in (**a**) a short-term test and (**b**) a long-term test as $I_S = 100$ mA.

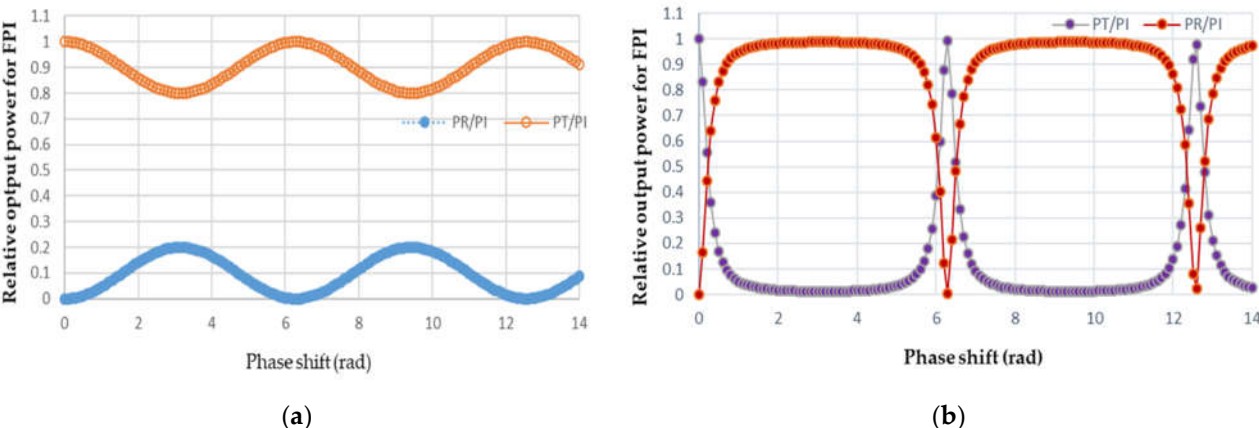

(a)　　　　　　　　　　　　　　　　　　　　　　　　　　(b)

**Figure 7.** Relative output power vs. phase shift in the cavity of a single FFPI with a mirror reflectance of (**a**) $R = 0.05$ and (**b**) $R = 0.8$, assuming no optical loss.

Within the adequate adjustment of the mismatch of these two cavity lengths with a TEC at the reference FFPI, there is no mismatch. Unfortunately, in this case, $\Delta L$ is approximately 8 μm, which is similar to the coherence length of the LED light source. The TEC at the reference FFPI provides compensation to correct for the mismatch and increase visibility, as shown in Figure 2. Whereas the optical path-length difference of the proposed low-cost sensing system is acceptably tolerated, the performance of temperature sensing is indeed close to that reported by Lee and colleagues [48].

Following all preliminary experiments, an acoustic modulator as a perturbed dynamic sinusoidal acoustic signal at a frequency of 1.74 Hz, as shown in Figure 8a, was implemented in the sensing FFPI. The sensing cavity produced a disturbing phase shift as a result of either a change in cavity medium index or cavity length modulation. The transmitted output signal at the reference FFPI, as depicted in Figure 8b, responded simultaneously and in accordance with the frequency of the acoustic stress source. With respect to the measured stress result, this acoustic wave sensor successfully detected the measurand. The total sensing wave form ($\Phi_T(t)$) shown in Figure 8b is composed of $\Phi_{sp}(t)$, $\Phi_{st}(t)$, and $\Phi_r(t)$,

indicating the signals from the speaker, Styrofoam sheet, and receiver, respectively. Because the acoustic wave in the Styrofoam sheet also generates some second-order disturbance in the sensing cavity, the entire waveform on the oscilloscope is distorted and not matched well with the original speaker signal. The signal noise on the waveform should chiefly come from the receiver. The frequency response is shown in Figure 9 in the frequency range of 0.1 to 200 Hz, indicating a cut-off frequency. The signal-to-noise ratio demonstrates a good performance at greater than 3 dB.

In recent years, reflective mirrors have been utilized to detect desired parameters [55]. Changes in $\Delta L_o$ due to disturbances in test circumstances impact the refractive index, cavity length, or both. The roundtrip phase shift in Equation (2) in the interferometers also varies and denotes the relationship between the phase shift and the measurand according to:

$$\Delta L_o = n \cdot \Delta L + L \cdot \Delta n \tag{10}$$

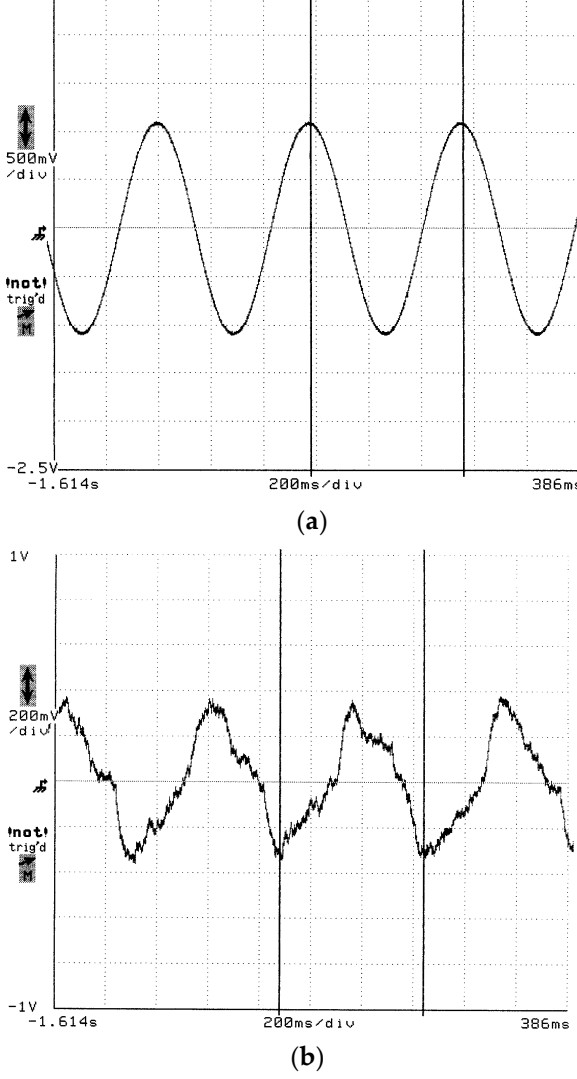

(**a**)

(**b**)

**Figure 8.** Oscilloscope traces showing the response of a sensor FFPI on a Styrofoam sheet: (**a**) disturbed signal applied to the speaker for modulation; (**b**) extracted signal from the optical receiver.

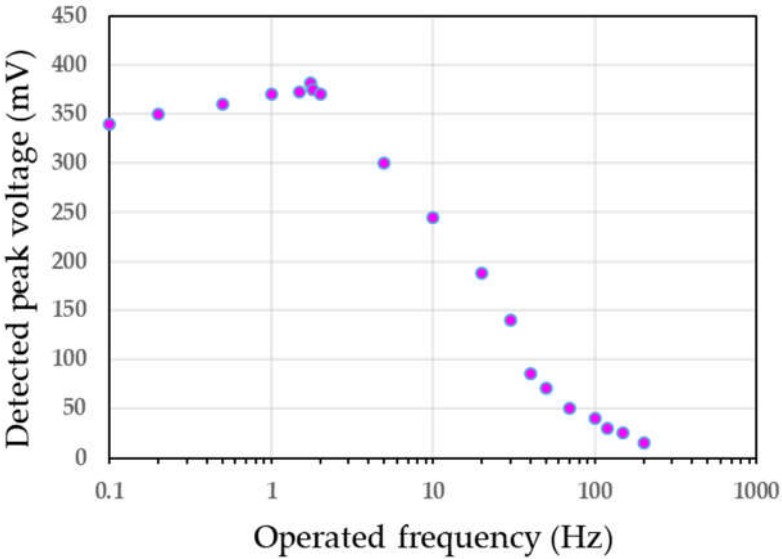

**Figure 9.** Frequency response of the acoustic vibration experiment from 0.1 to 200 Hz.

In general, the test circumstances occur not only in silica but also in fluid, air, or other objects. Thus, microelectromechanical systems, nanoelectromechanical systems, or other explicit cavity structures can be integrated into the sensing system to form more precise monitoring equipment in semiconductors or other advanced fields. In addition, to enhance the sensing performance, with an LED light source with a longer wavelength, i.e., 1.3 or 1.55 μm or shorter FWHM, Δλ is preferable to promote the coherence length. This action indirectly allows for prolonged mismatch of two FFPIs during manufacture. The success rate of a pair of FFPIs in manufacturing should be increased. The sensing experiment can likely be operated at room temperature without temperature compensation in the mismatch of cavity lengths. An alternative approach involves adjusting the reflectance of a sensing mirror, especially in bio applications [5,6,32]. However, this effect is mostly related to the variation of optical power intensity. If accuracy and multinode sensing are the main concerns, fiber Bragg grating technology can be integrated as a refractory optic system [56].

### 4. Conclusions

A low frequency-vibration sensor that integrates a dual FFPI sensing system with a low-cost and low-coherence LED light source is proposed to precisely detect acoustic vibrations. A novel technique for fabricating an intrinsic fiberoptic sensor with an optical path length greater than the coherence length of the LED light source was demonstrated. The dual FFPI sensing system consists of two Fabry–Perot interferometers. A sensing FFPI modulates the spectrum of LED emission, and a reference FFPI demodulates it. The characteristics of the components in the sensing system were investigated. Analytical and experimental results were obtained.

The experimental outcomes show that the dynamic measured consequences respond well in the dual FFPI to detect acoustic wave signals induced by vibration in a Styrofoam sheet. The sensing system demonstrates the high sensitivity of the optical receiver operating under sub-nW optical power. The development of feedback stabilization of a dual FFPI sensing system ensures optimal performance. In future research, the sensitivity and precision of the dual FFPI sensing system can be improved by using a longer peak wavelength, a shorter FWHM light source, or by adopting FBG technology to form the internal mirrors to achieve zero mismatch with respect to difference of cavity lengths. The proposed design using FBG technology is expected to provide strength, uniformity, and high reliability for the optical cavity of the FFPI, although cost and system complexity will be relatively increased.

**Author Contributions:** Conceptualization, M.-C.W.; methodology, S.-Y.C. and C.-Y.L.; validation, C.-Y.L., C.-H.-T.C. and W.-H.L.; formal analysis, all authors; investigation, M.-C.W.; data curation, S.-Y.C. and C.-H.-T.C.; writing—original draft preparation, M.-C.W.; writing—review and editing, all authors. All authors have read and agreed to the published version of the manuscript.

**Funding:** The authors sincerely appreciate the financial support from Ministry of Sci-ence and Technology of Republic of China under Contract MOST 110-2622-E-159-006-CC2.

**Conflicts of Interest:** The authors declare no conflict of interest.

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
