# Peer review of "Low-Frequency Vibration Sensor with Dual-Fiber Fabry–Perot Interferometer Using a Low-Coherence LED"

_crystals, doi:10.3390/cryst12081079_

Round 1

Reviewer 2 Report

Manuscript No:  crystals-1827996

Title:  Low-Frequency Vibration Sensor with Dual Fiber Fabry-Perot Interferometer Using a Low-Coherence LED Light Source

Authors:  Mu-Chun Wang, Shou-Yen Chao, Chun-Yeon Lin, Cheng-Hsun-Tony Chang and Wen-How Lan

A. Overview

1. In this manuscript the authors report on a Fabry–Perot interferometer for vibration sensing made off with a dual fiber and a low-coherence led light source.

2. The contents are expressed clearly.

3. The manuscript is well organized,

4. It is written in reasonable English.

5. The authors have acknowledged recent related research.

6. As long as my knowledge, the work presented is original.

7. Several typos along de manuscript, authors must have a second read

B. Detailed analysis.

Abstract is not clear enough.

-Please organize the ideas in each paragraph.

-Be clear, objective.

-State briefly what you did, how did you do it, the quantitative results you and

-State clearly the novelty of your work.

1. Introduction: provides an interesting approach to the subject and there are up to date references.

2. Sensing Principles of a Dual FFPI and Experimental Setup: provides the basic principles of Fabry-Perot interferometry – this can be shortened as it is well known.

There are too many figures in the manuscript which is unfavorable to reading.

For instance, figures 3 b and can be deleted.

Figure 4 is standard measurement technique, it can be withdrawn.

Figure 5 is LED characterization – not relevant to communicate new result to readers.

It is interesting that authors did a temperature characterization, but figure 6 is not needed – no new information.

C. Overall assessment

The work here presented is interesting. However it is too lengthy and too much information is presented, of which much is unnecessary. In my opinion the work cannot be published as is and must be re-written.

D. Review Criteria

1. Scope of Journal

Rating: Low

2. Novelty and Impact

Rating: Low

3. Technical Content

Rating: Low

4. Presentation Quality

Rating: Low

Author Response

Please see the attached file. Many thanks.

Round 2

Reviewer 1 Report

The revision has addressed all my concerns. I think the current version can be accepted for publication.

Reviewer 2 Report

The authors introduced corrections to the manuscript

and improved it. As well they answered

to the questions and queries of the reviewers.

In my opinion the ms can be published.